# Relevance of Early Introduction of Cow’s Milk Proteins for Prevention of Cow’s Milk Allergy

**DOI:** 10.3390/nu14132659

**Published:** 2022-06-27

**Authors:** Laurien Ulfman, Angela Tsuang, Aline B. Sprikkelman, Anne Goh, R. J. Joost van Neerven

**Affiliations:** 1FrieslandCampina, 3818 LA Amersfoort, The Netherlands; laurien.ulfman@frieslandcampina.com; 2Division of Allergy & Immunology, Department of Pediatrics, Icahn School of Medicine at Mount Sinai, New York, NY 10029, USA; angela.tsuang@mountsinai.org; 3Department of Pediatric Pulmonology and Pediatric Allergology, University Medical Center Groningen, University of Groningen, 9713 GZ Groningen, The Netherlands; a.b.sprikkelman@umcg.nl; 4University Medical Center Groningen, GRIAC Research Institute, University of Groningen, 9713 RB Groningen, The Netherlands; 5Department of Paediatrics, KK Women’s and Children’s Hospital, Singapore 229899, Singapore; anne.goh.e.n@singhealth.com.sg; 6Cell Biology and Immunology, Wageningen University, 6700 AH Wageningen, The Netherlands

**Keywords:** early introduction, food allergy, milk protein, prevention, cow’s milk allergy (CMA)

## Abstract

Food allergy incidence has increased worldwide over the last 20 years. For prevention of food allergy, current guidelines do not recommend delaying the introduction of allergenic foods. Several groundbreaking studies, such as the Learning Early About Peanut Allergy study, showed that the relatively early introduction of this allergenic food between 4–6 months of age reduces the risk of peanut allergy. However, less is known about the introduction of cow’s milk, as many children already receive cow’s-milk-based formula much earlier in life. This can be regular cow’s milk formula with intact milk proteins or hydrolyzed formulas. Several recent studies have investigated the effects of early introduction of cow’s-milk-based formulas with intact milk proteins on the development of cow’s milk allergy while breastfeeding. These studies suggest that depending on the time of introduction and the duration of administration of cow’s milk, the risk of cow’s milk allergy can be reduced (early introduction) or increased (very early introduction followed by discontinuation). The aim of this narrative review is to summarize these studies and to discuss the impact of early introduction of intact cow’s milk protein—as well as hydrolyzed milk protein formulas—and the development of tolerance versus allergy towards cow’s milk proteins.

## 1. Introduction

Food allergy is an immune reaction to the ingestion of food that is either IgE-mediated or non-IgE-mediated. The IgE-mediated allergic reaction is best understood and develops as a type 1 allergic reaction [1]. In food-allergen-sensitized individuals, the allergen induces an immediate response, typically within the first 2 h, leading to clinical symptoms that may involve the following systems: gastrointestinal, skin, respiratory, and cardiovascular [2]. Allergic reactions can occur to over 100 different foods, but egg, milk, and peanut allergens are the most common among children in Western countries, with egg and milk being particularly common under one year of age [3]. In analogy with the increased prevalence of asthma [4], the incidence in food allergies has increased considerably over the last few decades [5,6,7,8]. This is reflected by the atopic march theory that states that allergic diseases start in infancy with atopic dermatitis and food allergy and later progress to the development of allergic asthma and allergic rhinitis [9]. Allergic diseases are strongly associated with Westernized lifestyle and feeding practices, as prevalence in developing countries is also rising [10].

One of the key questions in relation to food allergy is when potentially allergenic foods should be introduced into the diet of infants to induce tolerance. As a result of several intervention trials on early introduction of food allergens, such as peanut and egg, there is increasing scientific consensus that earlier introduction of these foods can be recommended [11].

However, compared to other food allergens, the situation is quite different for the introduction of cow’s milk, as infants may in practice—when not exclusively breastfed—be exposed to cow’s milk proteins in cow’s-milk-based formula (CMF) at any time before 4 months and sometimes already in the first days of their life. Moreover, infants can also be exposed to low amounts of milk and other food proteins through breastmilk [12,13,14,15,16,17,18]. A large proportion of exclusively breastfed infants will not be introduced to dairy until they are weaned from breastfeeding. The most common allergens in cow’s milk are beta-lactoglobulin and caseins (beta, alpha s1 and s2, and kappa). The high prevalence of allergic patients with beta-lactoglobulin-specific IgE is explained by the absence of this protein in human milk. However, other whey proteins (bovine alpha-lactalbumin and bovine serum albumin (BSA)) have also been described as cow’s milk allergens [19]. 

Recent insights have increased our knowledge about the relationship between the timing and dose of cow’s milk exposure and the development of cow’s milk allergy (CMA). Therefore, the aim of this review is to summarize what is known about the timing of introduction of cow’s milk protein in early life and its influence on the development of tolerance versus allergy. We will discriminate between very early (first days of life) and early (first weeks to months of life) introduction and discuss the role of hydrolysates in the prevention of cow’s milk allergy.

## 2. Early Introduction of Food Allergens

For many years, dietary guidelines to prevent food allergy recommended a delayed introduction of food allergens into the diet of infants. This has changed in recent years following the publication of several studies that indicated that early introduction of food allergens may actually help to induce tolerance and thus prevent the development of food allergy, specifically for peanut and egg [20,21,22,23]. In addition, several studies have shown that the diversity of the diet in the first year of life is also associated with a lower incidence of food allergies [24]. The first insights for a tolerance-inducing effect when exposed to allergens early in life came from observational studies. It was found that the prevalence of peanut allergy among Jewish children in the UK was much higher compared to Israel, and the main difference was that a much higher percentage of children had ingested peanut protein in their diets by the age of 9 months in Israel compared to the UK [25]. This observation led to the Learning Early About Peanut Allergy (LEAP) study, in which the development of peanut allergy was studied in high-risk infants with eczema with or without egg allergy. The infants were enrolled between 4 to 11 months of age and randomized to avoid or consume peanut at a dose of 6 g of peanut protein/week. This study clearly demonstrated that the risk of peanut allergy at 5 years of age was significantly lower in the group that consumed peanuts (3.4%) compared to the avoidance group (20.3%) (ITT, 95% CI; 3.4 to 20.3; *p* < 0.001) [20,26]. The subsequent Enquiring About Tolerance (EAT) Study evaluated the effects of early introduction of six foods (peanut, egg, cow’s milk, sesame, whitefish, and wheat) in infants between 3 to 6 months of age alongside continued breastfeeding in comparison to exclusive breastfeeding until 6 months of age. The EAT study reported a significant decrease in the number of children from a general population developing peanut and egg allergy by 3 years of age in the per protocol analysis despite reported difficulties in adherence to the protocol [22]. However, the rates of other food allergies, including milk allergy, were not high enough to show significant effects. Nevertheless, the average relative risk of a positive skin-prick test at the age of 36 months to six food allergens was 79% lower in the early-introduction group compared to the standard food introduction group; These findings reached significance for peanut (*p* = 0.007), milk (*p* = 0.02), and sesame (*p* = 0.04). In addition to the EAT study, there are several randomized controlled trials that have investigated the effects of early egg introduction (STAR [27], HEALTHNUTS [28], STEP [29], HEAP [30], EAT [22], BEAT [31], PETIT [32]). A meta-analysis by Ierodiakonou [21] included five trials (1915 participants) and concluded that there was evidence with moderate certainty that introduction of egg at 4 to 6 months was associated with a lower egg allergy risk (risk ratio [RR], 0.56; 95% CI, 0.36–0.87; *I*^2^ = 36%; *p* = 0.009). Although a recent pilot study suggested that the early introduction of mixtures of many allergenic foods may be safe and efficacious for preventing food allergy [33], the practicality and safety of this approach should be carefully evaluated in future studies with such products. To prevent food allergy, the dosage of allergens should be sufficient and has been reported to be too low in some other commercial preparations [34]. Furthermore, a disadvantage of providing a mixture of multiple allergens is that when an allergic reaction occurs, it requires additional investigations to ascertain which allergen was responsible for the reaction. It is not clear if such a reaction to one of the allergens would also affect responses to other allergens in the mix.

As a result of all these new insights, guidelines have been updated in many different countries. These guidelines have been reviewed in a systematic review by Vale et al [11]. These changes are also illustrated in the policy document of the American Academy of Pediatrics [35,36] that shifted from the statement that “there are insufficient data to document a protective effect of any dietary intervention beyond 4 to 6 months of age for the development of atopic disease” in 2008 [35] to “There is no evidence that delaying the introduction of allergenic foods, including peanuts, eggs, and fish, beyond 4 to 6 months prevents atopic disease. There is now evidence that early introduction of peanuts may prevent peanut allergy” in 2019 [36]. In line with this, Fleischer et al. [37] recently published a consensus document with guidance to prevent food allergy on behalf of the American Academy of Allergy, Asthma & Immunology (AAAAI), American College of Allergy, Asthma & Immunology (ACAAI), and the Canadian Society of Allergy and Clinical Immunology (CSACI). Here, it was stated “to prevent peanut and/or egg allergy, both peanut and egg should be introduced around 6 months of life, but not before 4 months” and “other allergens should be introduced around this time as well. “Also, in other parts of the world, guidelines have been renewed [38] with careful guidance on the early introduction of allergenic food (egg, peanut) in high-risk infants (British Society for Allergy and Clinical Immunology (BSACI) 2018 [39], European Society for Paediatric Gastroenterology Hepatology and Nutrition (ESPGHAN) 2017 [40], National Institute of Allergy and Infectious Diseases (NIAID) 2017 [41]. The most recent European Academy of Allergy and Clinical Immunology (EAACI) guideline [42] not only provides guidance on egg and peanut introduction but also cow’s milk introduction (as discussed below).

## 3. Early Introduction of Cow’s Milk and Development of Cow’s Milk Allergy

Multiple studies have shown that introduction of cow’s milk protein in the first hours to days after birth followed by inconsistent incorporation into the diet is associated with increased risk of developing CMA (Table 1). Early studies showed that brief exposure to cows’ milk during the first three days of life in breast fed children was not associated with atopic disease or allergic symptoms up to age 5 [43,44]. However, cow’s milk allergy was not determined in these studies. Indeed, very early introduction of cow’s milk formula supplementation in the first 24 h of life increased the development of CMA in infants who were exclusively breastfed, as shown by a retrospective case–controlled study conducted in Ireland on 55 cow’s-milk-allergic infants born between 2010–2011 [45].

Using logistic regression, the only risk factor for developing CMA was formula supplementation in the first 24 h of life in exclusively breastfed infants (OR 7.01; 95%CI 1.79–27.01, *p* < 0.001). There was no increased risk of CMA in infants who were exclusively breastfed (without need for formula supplementation) or exclusively formula fed from birth onwards [45]. In another prospective study [46], it was found that infants that had been exclusively breastfed from birth and subsequently developed CMA had been supplemented with cow’s milk formula in the first 3 days of life. In another retrospective observational study in IgE-mediated CMA children, exposure to isolated doses of formula feeding in the hospital followed by exclusive breastfeeding was identified as a risk factor in the development of CMA [47].

Furthermore, a prospective study in more than 6000 infants [48] supported this finding by showing that infants who required supplementary feeding and received CMF while in the maternity hospital in the first 3 days of life had an increased risk for developing CMA (OR, 1.54; 95% CI, 1.04–2.30; *p* = 0.03) as compared to infants who received an extensively hydrolyzed whey formula (OR 0.61;95%CI, 0.38–1.00). Thus, preventing exposure to intact cow’s milk proteins through supplementation with extensively hydrolyzed cow’s milk devoid of allergenic proteins in the first 3 days of life may reduce the risk of developing CMA. Of interest is the observation that the infants that developed CMA in the breastfeeding group supplemented with CMF had a lower intake of formula in the first 8 weeks of life compared to the infants that did not develop CMA in that group [48]. The early exposure to cow’s milk followed by a period of avoidance of the cow’s-milk-based formula once breastfeeding is established likely leads to this increase in CMA. Based on these insights, EAACI recommended to “Avoid supplementing with cow’s milk formula in breastfed infants in the first week of life to prevent cow’s milk allergy in infants and young children” [42].

In contrast, observational studies have found that early introduction (after the first days of life but within the first weeks of life) of cow’s milk and continuation in the diet is associated with a lower chance of developing CMA [49,51,52], see Table 1. Thus, besides timing, the duration and frequency of allergen introduction during breastfeeding is of importance. The case–control study by Onizawa et al. demonstrated that early introduction of formula based on cow’s milk was associated with a decrease in the incidence of cow’s milk allergy of 51 IgE-mediated-cow’s-milk-allergic infants compared to 102 healthy controls [49]. There were significantly increased odds (aOR 23.4, 95%CI 5.39–104.52) of developing CMA when introduction was delayed (starting > 1 month after birth) or ingestion was irregular (less than once daily), further supporting the importance of duration and frequency. Likewise, using data from a large Japanese birth cohort involving over 100,000 mother–child pairs, analysis performed on 80,408 children showed that the regular consumption of cow’s milk based formula within the first 3 months of life was associated with a lower risk of CMA at 6 and 12 months (aRR 0.42, 95%CI 0.30–0.57 and aRR 0.44, 95%CI 0.38–0.51, respectively). [50]. Furthermore, prospective cohort studies from Melbourne, Australia [51] and Israel [52] also found evidence for an association between early introduction of cow’s milk before the third and first month of life and a reduced likelihood of developing CMA. In the Australian study, the early exposure to cow’s milk (before 3 months of life) in a predominantly breastfed group (87%) was associated with a significant risk reduction of cow’s milk allergy (aOR 0.31, 95% CI 0.10–0.91) at 12 months of age. In the Israeli cohort, the greatest risk of developing cow’s milk allergy was apparent in the group that introduced cow’s milk between 3.5 and 6.5 months of age after being exclusively breastfed. They observed that the risk of IgE-mediated CMA was lower in the infants of Arab women than Jewish women, and this difference was attributed to the decreased likelihood for Arab mothers to exclusively breastfeed their infants, instead providing cow’s-milk-based formula while breastfeeding. This finding supports the hypothesis that early and regular exposure to cow’s milk in the context of breastfeeding is important. In line with this, a retrospective study by Sakihara et al. [53] showed that continuous ingestion of cow’s milk reduced the risk of cow’s milk allergy in a population of infants with hen’s egg allergy based on positive oral food challenge. The infants were categorized into 4 groups: exclusively breast-fed, discontinuation of cow’s milk formula before the age of 3 months (temporary formula group), continuous but not daily ingestion of cow’s milk formula up to 3 months of age (non-daily group) and continuous ingestion of cow’s milk formula at least once daily (daily group). The non-daily group and daily group had significantly lower odds to develop CMA (OR 0.43, *p* = 0.02 and OR 0.11, *p* < 0.001, respectively) compared to the breastfeeding reference group, whereas the temporary group did not show a significant difference in CMA (OR 0.75, *p* = NS). However, a systematic review and meta-analysis by Ierodiakonou [21] showed that early introduction of cow’s milk did not significantly reduce the risk for development of CMA compared to late introduction (early *n* = 762 vs. late *n* = 788) and was based on two intervention studies [54,55]. A possible effect in the Lowe study [54] could have been missed due to the selection of infants (high risk) and/or the relative low number of infants that were exposed early to milk and/or avoided allergenic foods—including dairy—in the first year of life. In the study by Perkin et al. [55], the aim was to investigate the effect of early introduction of cow’s milk. The standard introduction group in the EAT study was allowed to supplement with CMF between 3 to 6 months of life if less than 300 mL, which already may have been enough to induce tolerance [55].

The best study design thus far to support early introduction and regular consumption of cow’s milk to prevent CMA is provided by the SPADE study (Strategy for prevention of milk allergy by daily ingestion of infant formula in early infancy) [57]. This prospective multicenter, open-label randomized controlled trial conducted in Okinawa, Japan included infants from the general population who were ingesting cow’s milk daily (ingestion group) from 1 to 2 months of age compared to an avoidance group that were supplemented with soy formula. Continuation of breastfeeding up to 3 and 6 months was high (89.5 vs. 89.7% and 72.2 vs. 67.7%, respectively, for ingestion vs. avoidance group). The ingestion group showed a significant reduction in CMA as diagnosed by open oral cow’s milk challenge at 6 months compared to the avoidance group (risk ratio = 0.12; 95% CI = 0.01–0.50; *p* < 0.001). Furthermore, infants in the avoidance group also showed higher sensitization to cow’s milk manifested as a positive skin prick test. Infants in the intervention group had higher casein-specific IgG4, supporting the hypothesis that tolerance induction is due to skewing of the immune response. The authors [57] concluded that cow’s milk formula should be started early, before the first month of life, and should be continued daily to reduce the risk of cow’s milk allergy while maintaining breastfeeding. The relatively high number of cow’s-milk-allergic-infants in the soy formula group may be the result of avoidance of cow’s milk allergen in the first 2 months of life. This is in line with earlier observations that avoidance may be detrimental [47]. Furthermore, in a subsequent subgroup analysis, Sakihara [57] showed that infants who received cow’s milk supplementation in the first 3 days of life who were subsequently randomized to the avoidance group also showed an increased risk of developing CMA. This is again in line with a difference in risk for CMA between very early introduction of cow’s milk in the first days of life followed by long duration of avoidance compared to early but continuous supplementation of cow’s milk from the first months of life. More observational and intervention studies are needed and are, according to clinicaltrials.gov, underway [58,59].

## 4. Prevention of Cow’s Milk Allergy: Hydrolysates

As discussed above, the very early introduction of cow’s milk in the first days of life followed by a period of avoidance is associated with increased risk of developing CMA. Therefore, it is important to know which supplements can be provided to support breastfeeding early in life without increasing this risk. Saarinen et al. [48] compared the supplementation of healthy infants during their stay in the maternity hospital with an extensively hydrolyzed whey formula, regular CMF, or pasteurized human milk to an exclusively breastfed group as the control. These infants were followed for 18 to 34 months for the development of CMA. In the group of infants receiving the whey hydrolysate, 1.5% of the infants developed cow’s milk allergy (OR 0.61; 95% CI, 0.38–1.00) compared to 2.4% in the group of infants receiving regular cow’s milk formula. In the infants receiving pasteurized human milk, 1.7% developed CMA (0.70; 95% CI, 0.44–1.12). These trends, however, did not reach statistical significance. In addition to early exposure to cow’s milk, parental history of allergy—defined as asthma, atopic dermatitis, allergic rhinitis, or conjunctivitis as determined by a questionnaire—also increased the risk of developing CMA in line with current knowledge on genetic risk. A randomized clinical trial [60] investigated the risk of sensitization towards cow’s milk and the development of allergic disease by early supplementation (in the first 3 days of life, but without continuous exposure) of breastfed infants with cow’s milk formula (CMF) or an amino-acid-based formula (AAF) in high-risk infants (defined as one immediate family member with atopic disease). Sensitization to cow’s milk (IgE level ≥ 0.35 allergen units [UA]/mL) occurred in 16.8% in the AAF group compared to 32.3% in the CMF group. Prevalence of food allergy at the second birthday was also significantly lower in the AAF group (2.6%) compared to the CMF group (13.2%) (RR 0.2; 95% CI, 0.07–0.57). The differences were not only present for cow’s-milk-induced allergic reactions but also egg- and wheat-related allergic reactions, which were increased in the CMF group. The mechanism for this is not clear. It may also be that those diagnosed with cow’s milk allergy are more likely to delay introduction of other allergenic foods, thereby missing the window of tolerance acquisition. Furthermore, the risk of asthma and recurrent wheeze was reduced in those who avoided early CMF when followed up to their second birthday [61]. The EAACI guidelines therefore suggest hydrolyzed formula as one of the possible temporary supplementary options depending on clinical, cultural, and economic factors [42].

The effectiveness of hydrolysates, either extensive or partial, on the reduction of cow’s milk allergy introduced later in life (beyond 14 days) is less clear. In fact, the conclusion of a recent EAACI guideline by Halken et al. [42] concluded that “There is no recommendation for or against using partially or extensively hydrolyzed formula to prevent CMA in infants. When exclusive breastfeeding is not possible many substitutes are available for families to choose from, including hydrolyzed formulas.” Halken et al. based this recommendation on nine trials [54,62,63,64,65,66,67,68,69] that either tested partially or extensively hydrolyzed casein or whey formulas on the occurrence of cow’s milk protein allergy. Cow’s milk allergy was diagnosed by means of clinical examination with either an open oral cow’s milk challenge which was followed by a double-blind challenge when the results were equivocal for the open challenge [65,68,69], single-blind challenge [66], double-blind challenge [62,63,67], clinical examination without an oral food challenge [64], or telephone interview [54]. In line with the EAACI statement, the recommendation from the AAAAI/ACAAI/CSACI Consensus Document [37] was “There is no protective benefit from the use of hydrolyzed formula in the first year of life against food allergy or food sensitization”. Furthermore, an earlier systematic review concluded that the use of hydrolyzed formula to prevent allergic disease in high-risk infants should not be recommended [70]. Many of the studies investigating the effect of hydrolyzed formula on the prevention of CMA have limitations which may have contributed to the lack of robust outcomes, such as (i) small sample size, (ii) design of the study, (iii) diagnostic criteria used, and (iv) characterization of the hydrolysate. With respect to the latter, peptide distribution, source of protein (casein and/or whey), and processing steps are thought to impact efficacy of hydrolysates [71,72,73]. Indeed, the lack of information on the distribution of peptides in a hydrolyzed formula was the main reason why the European Food Safety Authority (EFSA) recently concluded that a cause–effect relationship could not be established between the consumption of hydrolyzed infant formula and the risk reduction of atopic dermatitis [74]. Careful documentation and publication of the specifications of the hydrolysates used in clinical intervention studies will therefore be needed to overcome this shortcoming.

Thus, based on current literature, there is a role for hydrolysates in the first days of life in infants whose mothers are planning to subsequently breastfeed exclusively but cannot do so yet in the first days after giving birth. This should help to reduce early sensitization to cow’s milk protein. The role of continuous consumption of hydrolysates versus early cow’s milk formula consumption in the prevention of CMA is not so clear.

## 5. Limitations

A limitation of this narrative review is that it did not discuss other risk factors that may also play a role in development of CMA. These factors include, but are not limited to, preterm birth, the mode of delivery, and microbiota composition. Another limitation is that the review does not discuss non-IgE manifestations of CMA, and neither did it discuss the effect of combined preventive strategies.

## 6. Conclusions

The studies on early introduction of cow’s milk proteins into the infant diet via infant formulas have taught us that if mothers plan to exclusively breastfeed the infant, no cow’s milk formula should be given in the first weeks of life while breastfeeding is being established. If (donor) breast milk is not or insufficiently available in this very early period, extensively hydrolyzed milk formula or amino acid formula may be considered as an alternative [42].

Breastfeeding from birth with early introduction of cow’s milk supplementation within the first month of life and continued daily consumption of small amounts without hampering breastfeeding may reduce the risk of developing cow’s milk allergy. Additionally, the introduction of cow’s milk should not be followed by prolonged periods of avoidance, as this seems to increase the risk of developing cow’s milk allergy. Finally, even though many studies have been performed to date, there is currently no strong evidence that supports the use of hydrolyzed formula after 2 weeks of life for the prevention of CMA, indicating the need for additional studies with specific attention to the number of infants in the study, better characterization of the hydrolysates and their properties, and robust study design.

## Figures and Tables

**Table 1 nutrients-14-02659-t001:** Studies on very early introduction of cow’s milk in first days of life [45,46,47,48,49] and early introduction of cow’s milk [50,51,52,53,54,55,56,57].

Study	Design	Population	Infants/ Children	Age Onset Introduction CM	CM Formula	Breastfeeding	CMA Diagnosis	Outcome
Kelly et al. [45] 2019	Retro- and prospective	Infants at risk	55 infants	Within 24 h after birth	Regular CMF	BF only vs. BF + CMF	Allergy-focused clinical history, SPT, SpIgE measurement, and open food challenge, if necessary	Increased risk when CMF supplementation in first 24 h (OR 7.01; 95CI 1.79 27.01, *p* < 0.001)
Host et al. [46] 1988	Prospective	General population	1749 infants	Within first 3 days after birth	Regular CMF	BF population +/− early introduction of CMF in nursery	IgE and non-IgE, elimination/challenge test	39/1539 infants that received supplementation with CMF in the first 3 days had confirmed CMA while none of 210 exclusively BF neonates developed CMA (0/210), *p* < 0.05
Gil et al [47] 2017	Retrospective	CMA + infants	211 infants/group	Diverse	Regular CMF	Study focused on duration of IgE	IgE + CMA cases by clinical examination, provocation tests, serology	Increased risk when CMF supplementation in hospital, BF duration < 1 mo and 4–6 mo associated with higher risk of CMA while no increased risk of BF duration of 1–3 mo
Saarinen [48] 1999	Prospective	General population	6209 infants	Within first 20 h of life, and average feeding time of 2 days after birth	Pasteurized breastmilk Regular CMF Ext.Hydrolyzed whey formula	BF population (exclusively or supplemented with CMF, Hydrolyzed or pasteurized breastmilk)	Interview, elimination/challenge test, SPT	Feeding of CM at maternity hospitals increases the risk of CMA when compared with feeding of other supplements, but exclusive breast-feeding does not eliminate the risk
Sakihara [49] 2022	Data from SPADE study. Randomized controlled trial	Participants who ingested CMF in the first 3 days of life	431 children	4 groups of breastfed infants who discontinued CMF ingestion before age 1 month (“DISC < 1-month group”), during age 1 to 2 months (“DISC 1-2-month group”), during age 3 to 5 months (“DISC 3-5-month group”) not until age 6 months ("continuous group")	Mixed feeding groups (breastfeeding and cow’s milk formula (CMF) who discontinued CMF at different ages)	Breastfeeding + CMF in first 3 days of life +/− continuous CMF supplementation	Oral food challenge was performed to assess CMA development	CMA incidence was significantly higher in the DISC < 1 month group (*n* = 7 of 17, 41.2%; RR, 65.7; 95% CI, 14.7–292.5; *p* < 0.001), DISC 1–2-month group (*n* = 3 of 26, 11.5%; RR, 18.4; 95% CI, 3.2–105.3; *p* = 0.003), and DISC 3–5-month group (n = 7 of 69, 10.1%; RR, 16.2; 95% CI, 3.4–76.2; *p* < 0.001) than in the continuous group (*n* = 2 of 319, 0.6%)
Tezuka [50] 2020	Prospective	General population	>80,000 children	CMF consumption was categorized in < 3 mo, 3–6 mo or 6–12 mo at introduction	Regular CMF	BF and mixed fed.	CMA was defined as an allergic reaction to a CM product in an individual not consuming CM products at the time of evaluation, combined with physician-diagnosed food allergy	Introducing regular consumption of formula within the first 3 months of age was associated with lower risk of CMA at 12 months. Regular consumption at 3–6 months was strongly associated with a reduction in 12-month CMA (adjusted relative risks [95% confidence intervals]: 0.22 [0.12–0.35]), whereas no association was observed at 0–3 months (1.07 [0.90–1.27]
Peters [51] 2018	Longitudinal	General population	5276	Exposed to CMF 0–3 months or not	Regular CMF	Excl BF, mixed feeding, excl FF	Parental report of a reaction to cow’s milk consistent with IgE-mediated symptoms and a positive cow’s milk skin prick test	Early exposure to cow’s milk protein was associated with a reduced risk of cow’s milk sensitization (adjusted odds ratio [aOR] 0.44, 95% confidence interval [CI] 0.23–0.83), parent-reported reactions to cow’s milk (aOR 0.44, 95% CI 0.29–0.67), and cow’s milk allergy (aOR 0.31, 95% CI 0.10–0.91) at age 12 months
Katz [52] 2010	Prospective	General population	13019	Age at CMF exposure 0–14 days 15–104 days 105–194 days 195–374 days	Regular CMF	Excl BF, mixed feeding, excl FF feeding	Interview followed by SPT and open food challenge	The mean age of cow’s milk protein (CMP) introduction was significantly different (*p* < 0.001) between the healthy infants (61.6 ± 92.5 days) and those with IgE-mediated CMA (116.1 ± 64.9 days). Only 0.05% of the infants who were started on regular CMP formula within the first 14 days versus 1.75% who were started on formula between the ages of 105 and 194 days had IgE-mediated CMA (*p* < 0.001). The odds ratio was 19.3 (95% CI, 6.0–62.1) for development of IgE-mediated CMA among infants with exposure to CMP at the age of 15 days or more (*p* < 0.001)
Sakihara [53] 2016	Prospective	Hen’s-egg-allergic patients	397, <6 years of age	Excl BF group , discont ingestion of CMF before 3 mo of age (temp group; continuous ingestion of CMF, but not daily, up to 3 months of age (nondaily group); continuous ingestion of CMF at least once daily (daily group)	Regular CMF	Excl BF and mixed feeding groups	(1) a positive OFC result or any convincing episode of immediate reaction within 2 h after the ingestion of a cow’s milk product and [2] positive cow’s milk-specific IgE (CM-sIgE, > 0.34 KUA/L)	The incidence of developing CMA between the breast-fed group and temporary group did not show any statistical difference. Nondaily group and daily group had significantly lower incidence of developing CMA in comparison to the breast-fed group (nondaily group odds ratio 0.43; *p* = 0.02, daily group odds ratio 0.11; *p* < 0.001).
Lowe [54] 2011	Single-blind (participant) randomized controlled trial	Children with a family history of allergic disease	620, 0–2 y old children. Follow up at 6–7 years	randomized to receive the allocated formula at cessation of breast-feeding	cow’s milk formula, a pHWF, or a soy formula (after cessation of breastfeeding)	breast-feeding until cessation, followed by formula (cow’s milk formula, a pHWF, or a soy formula)	Skin prick tests to 6 common allergens (milk, egg, peanut, dust mite, rye grass, and cat dander) were performed at 6, 12, and 24 months.	The primary outcome was any allergic manifestation (cumulative incidence) up to 2 years of age. There was no evidence that infants allocated to the pHWF (odds ratio, 1.21; 95% CI, 0.81–1.80) or the soy formula (odds ratio, 1.26; 95% CI, 0.84–1.88) were at a lower risk of allergic manifestations in infancy compared with conventional formula. There was also no evidence of reduced risk of skin prick test reactivity or childhood allergic disease.
Perkin [55] 2016	Randomized controlled study	Exclusively breast-fed infants who were 3 months of age	1303 exclusively breast-fed infants randomized to (a) early introduction group of six allergenic foods (b) standard introduction group	As of 3 months of age for early introduction and as of 6 months for standard introduction group	peanut, cooked egg, cow’s milk, sesame, whitefish, and wheat	In the standard-introduction group, there was no consumption of peanut, egg, sesame, fish, or wheat before 5 months of age and consumption of less than 300 mL per day of formula milk between 3 and 6 months of age	Double blind placebo controlled food challenges, skin prick testing	The primary outcome was challenge-proven food allergy to one or more of the six early-introduction foods between 1 year and 3 years of age. In the intention-to-treat analysis, no significant differences were found. In the per-protocol analysis, the prevalence of any food allergy was significantly lower in the early-introduction group than in the standard introduction group (2.4% vs. 7.3%, *p* = 0.01), as was the prevalence of peanut allergy (0% vs. 2.5%, *p* = 0.003) and egg allergy (1.4% vs. 5.5%, *p* = 0.009); there were no significant effects with respect to milk, sesame, fish, or wheat. The early introduction of all six foods was not easily achieved but was safe
Sakihara [56] 2021	Randomized controlled trial (SPADE study)	Breastfed infants who (a) ingested at least 10 mL of CMF daily (b) avoided CMF but were given soy formula if needed	491 participants (242 in the ingestion group and 249 in the avoidance group)	Start CMF between 1 and 2 months of age	ingest at least 10 mL of CMF daily (ingestion group)	Breastfeeding +/− CMF or soy formula	Oral food challenge was performed to assess CMA development, skin prick test, serum titers specific IgE and IgG4	Primary outcome was CMA by oral food challenge. Secondary outcomes were proportion of infants with positive SPT and seruvm titers of specific IgE and IgG4. There were 2 CMA cases (0.8%) among the 242 members of the ingestion group and 17 CMA cases (6.8%) among the 249 participants in the avoidance group (risk ratio = 0.12; 95% CI = 0.01–0.50; *p* < 0.001). The risk difference was 6.0% (95% CI = 2.7–9.3). Approximately 70% of the participants in both groups were still being breast-fed at 6 months of age.Of the 227 ingestion group participants, 11 (4.8%) had a positive SPT response to cow’s milk at 6 months of age, as did 38 (16.2%) of the 235 avoidance group participants (RR 0.26; 95% CI 0.12–0.55; *p* < 0.001). The median titer of casein-specific IgG4 was 2.61 mgA/L (range, 0.45–10.46 mgA/L) in the ingestion group and 0.12 mgA/L (range, 0.08–0.33 mgA/L) in the avoidance group (P 0.02). Specific IgE titers did not significantly differ between the groups.
Onizawa [57] 2016	Retrospectively	CMA-allergic patients and non-allergic controls	51 IgE-CMA, 102 controls, 32 unmatched patients IgE egg. Over 1 year of age	Supplemented with CMF maternity clinic, excl BF, early regular CMF, delayed CMF, no early regular continuous CMF	Regular CMF	BF and mixed feeding population	Immediate allergic reactions, CM specific IgE (≥0.7 kUA/L), doctors diagnose of allergy	In a multivariable logistic regression analysis, the adjusted odds ratio of delayed (started more than 1 month after birth) or no regular cow’s milk formula (less than once daily) was 23.74 (95% CI, 5.39–104.52) comparing the CMA group with the Control group

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
