# Peer review of "Relevance of Early Introduction of Cow’s Milk Proteins for Prevention of Cow’s Milk Allergy"

_nutrients, 2022, doi:10.3390/nu14132659_

Round 1

Reviewer 1 Report

The manuscript reviewed the relationship between early introduction of cow milk protein and cow milk allergy.  However, the manuscript did not use a systematic way to review the topic. The detail information about obtaining references was not available. 

In addition, hydrolyzation is a very  broad topic and the current manuscript may not cover it enough.

Author Response

Reviewer 1

The manuscript reviewed the relationship between early introduction of cow milk protein and cow milk allergy.  However, the manuscript did not use a systematic way to review the topic. The detail information about obtaining references was not available. 

In addition, hydrolyzation is a very  broad topic and the current manuscript may not cover it enough.

Replies to Reviewer 1:

We thank the reviewer for the comments. We agree that this is not a systematic review. We regret it if this was somehow suggested. To make this clear we had already included the phrase  “ narrative review” in the last paragraph of the abstract (line 33 of revised version).

We agree with the reviewer that hydrolysation is a very broad topic. We would like to stress that studies that tested the effect of hydrolysates on atopic dermatitis were not the scope for the review. As with the other studies on intake of milk proteins, we considered only those that had an endpoint on cow’s milk protein allergy. This was a deliberate choice since many reviews, including meta-analysis papers on the effect of hydrolysates on prevention of atopic dermatitis have been published previously, we did not want to repeat that topic. We wanted to focus on the effect of milk proteins and hydrolysates on the development of cow’s milk allergy.

Reviewer 2 Report

I have read the review article 'Relevance of early introduction of cow’s milk proteins for prevention of cow’s milk allergy' of Ulfman and co-authors with much interest. It is a very clear and comprehensive review and useful for the decision of early introduction of cow's milk in an infants diet. Although most relevant studies have been incorporated, I would like to ask the authors to add 4 more refs to make the review complete with all existing data on early introduction of cow's milk. These are:

Gil, F. Association between caesarean delivery and isolated doses of formula feeding in cow milk allergy. Int Arch Allergy Immunol 2017 173:147-152

Tezuka, J. Possible association between early formula and reduced risk of cow's milk allergy: The Japan Environment and Children's Study. Clin Exp Allergy 2021 51:99-107

de Jong, M.H. Randomised controlled trial of brief neonatal exposure to cows' milk on the development of atopy. Arch Dis Child 1998 79:126-130

de Jong, M.H. The effect of brief neonatal exposure to cows' milk on atopic symptoms up to age 5. Arch Dis Child 2002 86:365-369

Author Response

Reviewer 2

I have read the review article 'Relevance of early introduction of cow’s milk proteins for prevention of cow’s milk allergy' of Ulfman and co-authors with much interest. It is a very clear and comprehensive review and useful for the decision of early introduction of cow's milk in an infants diet. Although most relevant studies have been incorporated, I would like to ask the authors to add 4 more refs to make the review complete with all existing data on early introduction of cow's milk. These are:

Gil, F. Association between caesarean delivery and isolated doses of formula feeding in cow milk allergy. Int Arch Allergy Immunol 2017 173:147-152

Tezuka, J. Possible association between early formula and reduced risk of cow's milk allergy: The Japan Environment and Children's Study. Clin Exp Allergy 2021 51:99-107

de Jong, M.H. Randomised controlled trial of brief neonatal exposure to cows' milk on the development of atopy. Arch Dis Child 1998 79:126-130

de Jong, M.H. The effect of brief neonatal exposure to cows' milk on atopic symptoms up to age 5. Arch Dis Child 2002 86:365-369

Replies to Reviewer 2:

We thank the reviewer for the compliments and the suggestions for the suggested relevant papers.

We have included these four references now in the revised manuscript by inserting the following sections in the revised manuscript:

Lines 281-283 “Brief exposure to cows’ milk during the first three days of life in breast fed children was not associated with atopic disease or allergic symptoms up to age 5 (De Jong 1998, 2002). However, cow’s milk allergy was not determined in these studies.”

Lines 293-296: “In another retrospective observational study in IgE-mediated CMA children, exposure to isolated doses of formula feeding in the hospital followed by exclusive breastfeeding was identified as a risk factor in the development of CMA (Gil 2017). “

Lines 336-340: “Likewise, data from a large Japanese birth cohort involving over 100 000 mother-child pairs, analysis done on 80 408 children showed that the regular consumption of cow’s milk based formula within the first 3 months of life was associated with a lower risk of CMA at 6 and 12 months (aRR 0.42, 95%CI 0.30-0.57) and (aRR 0.44, 95%CI 0.38-0.51) respectively (Tezuka 2021).”

Reviewer 3 Report

This is a clear and well-written narrative review assessing the effect of early introduction of cow's milk on the development of cow's milk allergy. A section on current uncertainity, including possible determinants of outcomes and data on preterm infants, type of delivery, microbiota, non-IgE manifestations of CMA, effect of combined preventive strategies could be considered and added to the manuscript.

In the  conclusion the first and last part about the use of hydrolyzed formulas is a bit contradictory.

Please clarify

Author Response

Reviewer 3

This is a clear and well-written narrative review assessing the effect of early introduction of cow's milk on the development of cow's milk allergy. A section on current uncertainity, including possible determinants of outcomes and data on preterm infants, type of delivery, microbiota, non-IgE manifestations of CMA, effect of combined preventive strategies could be considered and added to the manuscript.

In the  conclusion the first and last part about the use of hydrolyzed formulas is a bit contradictory.

Please clarify

Replies to Reviewer 3:

We thank the reviewer for the kind remarks. We added to the discussion section a part on current uncertainties that were mentioned by the reviewer and this can be found in the second paragraph as well as the last paragraph (Lines 628-638):

Thus based on current literature, the role of hydolysates is in early life for supplementation whilst establishing breastfeeding in those infants where mothers are going to exclusively breastfeed subsequently to reduce early sensitization to cow’s milk protein. The role of continuous consumption of hydrolysates versus early cow’s milk formula consumption in the prevention of CMA is not so clear.

  1. Limitations

A limitation of this narrative review is that it did not discuss other risk factors that may also play a role in development of CMA. These factors include but are not limited by preterm birth, the mode of delivery and microbiota composition. Another limitation is that the review does not discuss non-IgE manifestations of CMA and neither did it discuss the effect of combined preventive strategies. “

Furthermore, we agree with the reviewer that the part on the use of hydrolysates is a bit contradictory. To prevent confusion we added the time window for which the use of hydrolysates is not clearly related to a preventive benefit. And that time window is after the 2 weeks of life. In other words, the first 3 days of life hydrolysates do prevent CMA compared to intact cow’ s milk formula, but introduction of hydrolysates compared to intact cow’s milk formula after 2 weeks is not clearly beneficial. We addressed this by adding the part in yellow we think this confusion is taken away:“ Finally, even though many studies have been performed to date, there is currently no strong evidence that supports the use of hydrolyzed formula, after 2 weeks of life, for the prevention of CMA” (line 655-658 of the revised manuscript).

Reviewer 4 Report

In the introduction you should add information which cow's milk proteins cause allergy. You can't generalize like that, because it may be an allergy to alpha-casein or some whey proteins. For research, it is easy to separate casein from whey proteins using membrane techniques. 

Authors should complete information on the recommendations of the World Allergy Organization (WAO). Published in 2022: Diagnosis and Rationale for Action against Cow's Milk Allergy (DRACMA) Guidelines update - I - Plan and definitions.

Remove  of keywords: Peanut; Egg

Another important aspect to account for is  effect of associating probiotics with formulas, either administered separately or mixed in the same formulation, on the duration of IgE-CMA. Another issue that should be described is the use of new formulas based on amino acids with synbiotic supplementation.

Author Response

Reviewer 4

In the introduction you should add information which cow's milk proteins cause allergy. You can't generalize like that, because it may be an allergy to alpha-casein or some whey proteins. For research, it is easy to separate casein from whey proteins using membrane techniques. 

Authors should complete information on the recommendations of the World Allergy Organization (WAO). Published in 2022: Diagnosis and Rationale for Action against Cow's Milk Allergy (DRACMA) Guidelines update - I - Plan and definitions.

Remove  of keywords: Peanut; Egg

Another important aspect to account for is  effect of associating probiotics with formulas, either administered separately or mixed in the same formulation, on the duration of IgE-CMA. Another issue that should be described is the use of new formulas based on amino acids with synbiotic supplementation.

Replies to Reviewer 4:

We thank the reviewer for the comments.

We added information about the type of allergens in cow’s milk to the Introduction : The most common allergens in cow’s milk are beta-lactoglobulin and caseins (beta, alpha s1 and -s2 and kappa). The high prevalence of allergic patients with beta lactoglobulin specific IgE is explained by the absence of this protein in human milk. Yet, other whey proteins (alpha-lactalbumin, bovine se-rum albumin (BSA) have also been described as cow’s milk allergens (Savilahti 1992). (Revised manuscript, Lines 62-66)

The DRACMA update of 2022 by Fiocchi et al which the reviewer mentions is an important document. However, this article does not discuss or review guidelines for timing of introduction of cow’s milk proteins or hydrolysates to prevent the development of cow’s milk allergy. For this reason, we have chosen not to include the reference to this paper in the current review.

The words peanut and egg have been removed from the keywords in the revised manuscript.

As to the suggestion of the reviewer to include probiotics and synbiotics and amino acid formula in the discussion, we respectfully disagree with the reviewer. The aim of this review was to focus on the intake of cow’s milk proteins or epitope-containing fragments thereof on the risk to develop cow’s milk allergy. The effects of other components like pre, pro, synbiotitcs, PUFA, vitamin D etc are of course very relevant but in our view out of scope in this specific review. We hope the reviewer can agree with this.

Reviewer 5 Report

This publication is a systematic review 11 studies on very early (in first days of life) or early  (between 4-6 months of age) introduction of cow’s milk based formulas with intact milk proteins or hydrolysed milk formulas on the development of cow’s milk allergy while breastfeeding.

I agree with the author’s conclusions that depending on the time of introduction and the duration of cow’s milk administration the risk of cow’s milk allergy can be reduced (early introduction) or increased (very early introduction) followed by discontinuation of breastfeeding.

The publication contains valuable indications on the appropriate time of introducing dairy products to the nutrition of newborns and infants as a preventive measure against the development of allergy to cow's milk proteins.

Author Response

Reviewer 5

This publication is a systematic review 11 studies on very early (in first days of life) or early  (between 4-6 months of age) introduction of cow’s milk based formulas with intact milk proteins or hydrolysed milk formulas on the development of cow’s milk allergy while breastfeeding.

I agree with the author’s conclusions that depending on the time of introduction and the duration of cow’s milk administration the risk of cow’s milk allergy can be reduced (early introduction) or increased (very early introduction) followed by discontinuation of breastfeeding.

The publication contains valuable indications on the appropriate time of introducing dairy products to the nutrition of newborns and infants as a preventive measure against the development of allergy to cow's milk proteins.

Replies to Reviewer 5:

We thank the reviewer for the comments and for reviewing our manuscript. We would like to make the point that it is not a systematic review, as also remarked in our comments to reviewer 1. To make this clear we have added “ narrative review” in the last paragraph of the abstract (line 33 of revised manuscript).

Round 2

Reviewer 1 Report

1.     Line 60 “alpha-lactalbumin” could be revised to “bovine alpha-lactalbumin”?

2.     Lines 64-65 “the aim of this review is to summarize what is known about the timing 64 of introduction of food allergens like egg and peanut”. Egg and peanut introduction could not be considered as the focus of the purpose of the study based on the title that is the review of early introduction of cow milk.

3.     “In a multivariable logistic regression anal-ysis, the adjusted odds ratio of delayed (started more than 1 month after birth) or no regular cow’s milk formula (less than once daily) was 23.74 (95% CI, 5.39-104.52) comparing the CMA group with the Con-trol group” the summary is not clear which group is control group and which is the exposure group. “ Onizawa (49) 2016”

4.      “ extensively hydrolyzed whey formula (OR 0.61;95%CI, 0.38-1.00).” Only one study showed that hydrolyzed whey formula may reduce the odds of CMA, which may not be the strong evidence to recommend to use either hydrolyzed or amino acid formula in the conclusion.

Author Response

  1. Line 60 “alpha-lactalbumin” could be revised to “bovine alpha-lactalbumin”?

This has been changed in the revised manuscript.

  1. Lines 64-65 “the aim of this review is to summarize what is known about the timing 64 of introduction of food allergens like egg and peanut”. Egg and peanut introduction could not be considered as the focus of the purpose of the study based on the title that is the review of early introduction of cow milk.

The sentence

“Therefore, the aim of this review is to summarize what is known about the timing of introduction of food allergens like egg and peanut, and especially what is known about intact cow’s milk protein in early life and its influence on the development of tolerance versus allergy.”

Has now been changed into:

“Therefore, the aim of this review is to summarize what is known about the timing of introduction of cow’s milk protein in early life and its influence on the development of tolerance versus allergy. “

  1. “In a multivariable logistic regression anal-ysis, the adjusted odds ratio of delayed (started more than 1 month after birth) or no regular cow’s milk formula (less than once daily) was 23.74 (95% CI, 5.39-104.52) comparing the CMA group with the Con-trol group” the summary is not clear which group is control group and which is the exposure group. “ Onizawa (49) 2016”

We checked the refernce, and it is indeed as stated above. The authors compared the OR of the children that had either delayed- or no regular cow’s milk formula with the control group for CMA (odds ration), so the comparison was between these two combined and the control group. Therefore we kept the text in the Table as it was.

  1. “ extensively hydrolyzed whey formula (OR 0.61;95%CI, 0.38-1.00).” Only one study showed that hydrolyzed whey formula may reduce the odds of CMA, which may not be the strong evidence to recommend to use either hydrolyzed or amino acid formula in the conclusion.

This section preceding the sentence was in the manuscript:

“In another prospective study (46) it was found that infants that had been exclusively breastfed from birth and subsequently developed CMA, had been supplemented with cow’s milk formula in the first 3 days of life. In another retrospective observational study in IgE -mediated CMA children, exposure to isolated doses of formula feeding in the hospital followed by exclusive breastfeeding was identi-fied as a risk factor in the development of  CMA (47).

Furthermore, a prospective study in more than 6000 infants (48) supported this finding by showing that infants who required supplementary feeding and received CMF while in the maternity hospital in the first 3 days of life, had an increased risk for developing CMA (OR, 1.54; 95% CI, 1.04-2.30; P = .03) as compared to infants who received an extensively hydrolyzed whey formula (OR 0.61;95%CI, 0.38-1.00).”

We agree with the reviewer that the following sentence “Thus, preventing exposure to intact cow’s milk proteins through supplementation with extensively hydrolyzed cow’s milk devoid of allergenic proteins in the first 3 days of life reduces the risk to develop CMA.” was worded too strongly.

We changed this sentence into:” Thus, preventing exposure to intact cow’s milk proteins through supplementation with extensively hydrolyzed cow’s milk devoid of allergenic proteins in the first 3 days of life may reduce the risk to develop CMA.”
